# A Multi-Faceted Analysis Showing *CRNDE* Transcripts and a Recently Confirmed Micropeptide as Important Players in Ovarian Carcinogenesis

**DOI:** 10.3390/ijms25084381

**Published:** 2024-04-16

**Authors:** Anna Balcerak, Laura Aleksandra Szafron, Tymon Rubel, Bianka Swiderska, Arkadiusz M. Bonna, Magdalena Konarzewska, Ireneusz Sołtyszewski, Jolanta Kupryjanczyk, Lukasz Michal Szafron

**Affiliations:** 1Department of Pathology and Anatomical Sciences, State University of New York, Buffalo, NY 14203, USA; 2Department of Molecular and Translational Oncology, Maria Sklodowska-Curie National Research Institute of Oncology, 02-781 Warsaw, Poland; 3Maria Sklodowska-Curie National Research Institute of Oncology, 02-781 Warsaw, Poland; 4Institute of Radioelectronics and Multimedia Technology, Warsaw University of Technology, 00-665 Warsaw, Poland; 5Mass Spectrometry Laboratory, Institute of Biochemistry and Biophysics, Polish Academy of Sciences, 02-106 Warsaw, Poland; 6Triple Helical Peptides Ltd., Cambridge CB22 5DU, UK; 7Department of Forensic Medicine, Warsaw Medical University, 02-007 Warsaw, Poland; 8Department of Cancer Pathomorphology, Maria Sklodowska-Curie National Research Institute of Oncology, 02-781 Warsaw, Poland

**Keywords:** ovarian cancer, lncRNA, microtubule, micropeptide, SEP, CRNDE, adhesion, nucleolus, centrosome, cytoskeleton, RNA-Seq, affinity chromatography, mass spectrometry

## Abstract

*CRNDE* is considered an oncogene expressed as long non-coding RNA. Our previous paper is the only one reporting *CRNDE* as a micropeptide-coding gene. The amino acid sequence of this micropeptide (CRNDEP) has recently been confirmed by other researchers. This study aimed at providing a mass spectrometry (MS)-based validation of the CRNDEP sequence and an investigation of how the differential expression of CRNDE(P) influences the metabolism and chemoresistance of ovarian cancer (OvCa) cells. We also assessed cellular localization changes of CRNDEP, looked for its protein partners, and bioinformatically evaluated its RNA-binding capacities. Herein, we detected most of the CRNDEP sequence by MS. Moreover, our results corroborated the oncogenic role of *CRNDE*, portraying it as the gene impacting carcinogenesis at the stages of DNA transcription and replication, affecting the RNA metabolism, and stimulating the cell cycle progression and proliferation, with CRNDEP being detected in the centrosomes of dividing cells. We also showed that CRNDEP is located in nucleoli and revealed interactions of this micropeptide with p54, an RNA helicase. Additionally, we proved that high CRNDE(P) expression increases the resistance of OvCa cells to treatment with microtubule-targeted cytostatics. Furthermore, altered CRNDE(P) expression affected the activity of the microtubular cytoskeleton and the formation of focal adhesion plaques. Finally, according to our in silico analyses, CRNDEP is likely capable of RNA binding. All these results contribute to a better understanding of the CRNDE(P) role in OvCa biology, which may potentially improve the screening, diagnosis, and treatment of this disease.

## 1. Introduction

The *CRNDE* (Colorectal Neoplasia Differentially Expressed) gene is generally considered an oncogene expressed as long non-coding RNA (lncRNA). Our team identified the first two complete *CRNDE* gene transcripts and submitted their sequences into the GenBank database, where they were assigned the FJ466685.1 and FJ466686.1 identifiers. The former transcript is lncRNA-coding only, while the latter encodes CRNDEP, an 84-amino-acid oncogenic micropeptide first described by our team in 2015 [1]. The FJ466685.1 transcript is most similar to the NR_034106.3 reference sequence, whereas the sequence of the FJ466686.1 variant most resembles the NR_170995.1 reference transcript of *CRNDE*. We also found that increased expression of both these transcripts was a negative prognostic factor in ovarian cancer (OvCa) patients. Moreover, the decreased expression of both *CRNDE* transcripts correlated with the accumulation of the TP53 protein in tumor cells [2]. Additionally, with the use of an antibody developed on our request, we found CRNDEP to be overexpressed in highly proliferating cells in normal and neoplastic tissues alike [1].

This study aimed to broaden the knowledge of the role of the *CRNDE* gene, its transcripts, and its protein product, CRNDEP, in cancer cell lines with different TP53 statuses. CRNDEP belongs to so-called smORF-encoded polypeptides (SEPs), i.e., peptides encoded by small open reading frames (smORFs). In recent years, three research teams independently confirmed that the *CRNDE* transcript (recognized by us as CRNDEP-coding) has the ability to bind ribosomes along its entire length. Moreover, the specificity of this binding indicated translation [3,4,5]. In addition, CRNDEP’s existence was recently confirmed by others in the mass spectrometry (MS) analysis [6]. The median length of SEPs identified so far is only 50 aa, which suggests that they probably do not exhibit enzymatic activity but only modulate the activity of proteins to which they bind or play the role of signaling molecules [7], such as myoregulin (MLN) and sarcolipin (SLN) [8]. Other studies have shown the impact of micropeptides on DNA repair, myogenesis, inflammation, metabolism, and carcinogenesis [6]. Herein, we concentrated on the validation of the CRNDEP amino acid sequence. Next, we investigated how the expression of *CRNDE* gene products influences the physiology and morphology of neoplastic cells. Additionally, our team assessed the impact of altered CRNDE(P) expression on the sensitivity of ovarian cancer cells with wild-type *TP53* or with a null mutation in this gene to cytostatics disturbing the microtubule metabolism. We also determined the cellular localization of the CRNDEP micropeptide and searched for its protein partners. Finally, we evaluated the RNA binding capacities of CRNDEP in silico.

## 2. Results

### 2.1. Establishment of Cell Lines Expressing the CRNDEP Micropeptide Fused to the Double Flag Tag

The expression cassettes harboring the puromycin resistance gene and open reading frames encoding either the Flag tag only or the Flag-CRNDEP fusion protein were both introduced to the AAVS1 safe harbor locus in the genomes of SK-OV-3 and A2780 cells by using the CRISPR/Cas9 technology. About 20 clones resistant to puromycin for each cell line were obtained. The results of Western/dot blot (for CRNDEP) and real-time quantitative PCR (RT-qPCR) (for the *CRNDE* transcripts) analyses, displaying how the expression of the relevant transgene changed in these clones, are shown in Appendix A. This approach led to the selection of two clones characterized by the highest CRNDE(P) expression levels for each cell line, named SK-OV-3-Flag-CRNDEP_K1, SK-OV-3-Flag-CRNDEP_K2, A2780-Flag-CRNDEP_K1, and A2780-Flag-CRNDEP_K2. These daughter cell lines were subjected to further investigation in this study as test samples. In addition, for each parent cell line, one control daughter cell line was developed, i.e., SK-OV-3-Flag_K1 and A2780-Flag_K1.

### 2.2. Development of SK-OV-3 Cell Lines with the CRNDE Gene Knockdown

First, the preliminary RT-qPCR analysis was performed in HeLa cells to determine which of three *CRNDE*-silencing short hairpin RNAs (shRNAs) (SH1, SH2, and SH3) elicits (after its ectopic, transient expression) the most significant decrease in the expression of *CRNDE* transcripts compared to the scrambled control hairpin, SH SCR. Noteworthy, the SH1 and SH2 hairpins are capable of knocking down all five reference *CRNDE* transcripts, i.e., *NR_034105.4*, *NR_034106.3*, *NR_110453.2*, *NR_110454.2*, and *NR_170995.1*. By contrast, the SH3 hairpin specifically diminishes the expression of the NR_170995.1 reference transcript only, the one encoding the CRNDEP micropeptide. This experiment proved the SH1 hairpin to silence the expression of *CRNDE* most efficiently, decreasing the expression of the CRNDEP-coding reference transcript by about 65% and other *CRNDE* reference splice variants by about 50% (Appendix A). Therefore, SH1 was selected for the establishment of two independent SK-OV-3 cell lines characterized by a stable knockdown of the *CRNDE* gene. Two independent control SK-OV-3 cell lines with the SH SCR hairpin integrated into their genomes were developed, as well. Ultimately, the *CRNDE* expression in all the daughter cell lines was evaluated by RT-qPCR, revealing at least a 75% decrease (for every transcript tested) in each cell line expressing the SH1 hairpin in comparison with the control cells (Appendix A).

### 2.3. Cell Cycle Phase-Dependent Changes in CRNDEP Localization

To investigate how the cellular localization of CRNDEP changes in different cell cycle phases, HeLa cells were cultured in media containing cell cycle inhibitors. In starving HeLa cells, stopped in the G0/G1 phase, CRNDEP exhibited nuclear localization (Figure 1A). When the cells were blocked in the G1/S phase, the concentration of CRNDEP within the granular structures in the nucleus was observed. The blockage of the cell cycle in the S phase led to the formation of nuclear ring-like structures abundant in CRNDEP. Interestingly, in dividing cells, CRNDEP migrated from the nucleus to centrosomes, which was further confirmed by the co-localization of CRNDEP with centrin, a marker of these organelles (Figure 1B).

### 2.4. Alterations in Cells’ Distribution between Different Cell Cycle Phases

The flow cytometry analysis (Figure 1C) revealed that the percentage of cells in the S phase of the cell cycle is significantly lower in SK-OV-3 cells with the shRNA-mediated silencing of *CRNDE* (29.0 ± 6.5%) than in the control cells (40.9 ± 4.1%; *p* = 0.0025). Consistently, the fraction of dividing cells (being in the G2/M phase) was also significantly lower after the *CRNDE* knockdown (0.5 ± 0.4%) compared to the control cells (1.4 ± 0.9%; *p* = 0.020).

### 2.5. Mitotic Index and Cell Viability Evaluation of the Cells with Different CRNDE(P) Levels

The mitotic index was assessed for SK-OV-3 and A2780 cells with stable overexpression of CRNDE(P) and also for SK-OV-3 cells after the stable CRNDE(P) knockdown. In both cell lines, the overexpression resulted in a significantly increased proliferation rate (Figure 2A,B). By contrast, cells with silenced CRNDE(P) expression divided less often than the control ones (Figure 2C). The MTT assay for cell viability assessment perfectly corroborated the above results, portraying the cells with the higher CRNDE(P) expression as more metabolically active than the control cells (Figure 2D,E). Consistently, CRNDE(P) knockdown elicited opposite metabolic effects (Figure 2F).

### 2.6. Cell Shape Changes Depending on CRNDE(P) Expression

To examine how alterations in the CRNDE(P) expression influence the cell shape changes, the cell shape index (CSI) value was calculated for SK-OV-3 cells with either stable knockdown or overexpression of CRNDE(P). After silencing, the cells became more elongated (median CSI ≈ 0.3) than in the control lines (median CSI ≈ 0.6), and this difference was statistically significant (Figure 3A,B,E,F). By contrast, the SK-OV-3 cells with CRNDE(P) overexpression exhibited no shape changes (Figure 3C,D,G,H).

### 2.7. CRNDE(P) Expression-Dependent Changes in the Speed of Microtubules’ Repolymerization

We found that the rate of microtubules’ repolymerization after the removal of nocodazole (NOC, microtubule polymerization inhibitor) from the culture medium was noticeably higher in SK-OV-3 cells with CRNDE(P) overexpression (SK-OV-3-Flag-CRNDEP_K1) than in the control cells (SK-OV-3-Flag_K1) (Figure 4).

### 2.8. CRNDE(P) Expression-Dependent Changes in Cell–Substrate Adhesion

When performing routine passages of the SK-OV-3 cell line, we observed a decreased substrate adhesion of the cells with CRNDE(P) knockdown. To further investigate the nature of this phenomenon, the number and the size of focal adhesion plaques (FAs) were assessed by immunodetecting paxillin, a marker of these structures. An exemplary, representative picture of FAs in SK-OV-3 cells with and without CRNDE(P) knockdown is shown in Figure 5A. With the use of the ImageJ software, the average number of FAs per cell and the mean size of a single FA plaque were calculated. Additionally, the average total area of FAs per cell was measured. The statistical analysis of the results revealed that the average number of FAs per cell (Figure 5B) and the total FA area per cell (Figure 5D) were both significantly diminished after CRNDE(P) silencing in comparison with the control cells. By contrast, the mean size of a single FA plaque (Figure 5C) did not depend on the CRNDE(P) expression. Thus, the CRNDE(P) knockdown leads to a decreased number of FAs in a cell, which in turn causes the reduction of a total FA area, resulting in the weakening of the cell–substrate adhesion strengths.

### 2.9. Cytotoxicity Tests

In our previous research [2], the *CRNDE* gene was identified as a negative prognostic factor in ovarian cancer patients who underwent the taxane/platinum (TP) treatment. However, this regularity was not found in the platinum/cyclophosphamide (PC)-treated patients. Therefore, in the present study, we carried out a detailed evaluation of how the altered CRNDE(P) expression affects the cellular response to various microtubule-targeted agents, i.e., NOC, vinblastine (VIN), paclitaxel (PTX), and noscapine (NOS). The results of the half-maximal inhibitory concentration (IC50) value assessment with the MTT test for each cell line/drug combination are presented in Figure 6A–D. Moreover, PTX and NOS were administered to the cells simultaneously to verify whether and how their synergistic therapeutic effect, discovered by other researchers [9], changes in OvCa cells with diverse CRNDE(P) expression (Figure 6E). The tests demonstrated that silencing of the *CRNDE* gene did not alter the sensitivity of the cells to any cytostatic compound used. On the contrary, CRNDE(P) overexpression increased the resistance of the cells to drugs, inhibiting microtubule polymerization (NOC, but not VIN) and interrupting their depolymerization (PTX). Such cells were also more resistant to NOS, which acts as a stabilizer for microtubules. Interestingly, the higher resistance of the cells to PTX, due to the elevated levels of CRNDE(P), may be overcome by the combined treatment with PTX and NOS. This favorable therapeutic outcome was found in A2780 and SK-OV-3 cells alike.

### 2.10. Assessment of Transcriptomic Alterations Caused by the CRNDE Gene Knockdown

Before the actual analysis of differential gene expression, our next-generation sequencing (NGS) data were thoroughly verified with a set of bioinformatic tools to ensure their reliability and minimize the chance of drawing false or inaccurate conclusions. First, the heatmaps showing the hierarchical clustering of the distances in overall gene expression patterns were generated to compare the similarities and differences between the SK-OV-3 cells with *CRNDE* knockdown and the control samples (Appendix A). This analysis revealed that transcriptomic changes among the clones of the same daughter cell line and their biological replicates were negligible compared to the expression alterations found between the cells with and without *CRNDE* knockdown. The overall transcriptomic patterns were also subjected to the principal component analysis (PCA), which revealed that the differences between the cells with *CRNDE* silencing and the control cells accounted for 91% of the total gene expression variance in this sample set (Appendix A). This proves the highest standards of cell culturing and all subsequent analytical steps, including RNA extraction, reverse transcription, NGS libraries’ preparation, and sequencing. To further increase the reliability of our in silico analysis and lower the risk of the type I statistical error (false positive), a comparison of four differentially expressed gene sets obtained with (sequence aligner+expression analyzer): HISAT2+edgeR, HISAT2+DESeq2, STAR+edgeR, STAR+DESeq2 was made (Appendix A). This analysis demonstrated that, out of 10,027 unique genes identified as differentially expressed in at least one set, 5934 genes (59.2%) were found in every set. This common set, consisting of 2939 down- and 2995 up-regulated genes (*CRNDE*-silenced vs. control group), was used in all downstream bioinformatic investigations described in this article. Interestingly, when comparing the genes with the most differentiated expression (200 with the lowest Benjamini–Hochberg correction (FDR) value, cutoff = 2.45 × 10^−76^, and 200 with the highest absolute log2FoldChange (log2FC) value, cutoff = 3.06) on a volcano plot (Appendix A), we discovered that among the genes meeting both conditions (N = 108), the number of up-regulated genes (N = 79) was significantly higher that the number of genes with the diminished expression after *CRNDE* knockdown (N = 29) (Chi−squared test *p*-value = 3.166 × 10^−6^).

Next, the gene ontology analysis was carried out for two sets of genes with altered expression (FDR < 0.05), being either down-regulated (fold change (FC) ≤ 0.8) or up-regulated (FC ≥ 1.2) due to *CRNDE* silencing. The same analysis was conducted for the joined sets, too. The results, presented in the Appendix A, revealed that *CRNDE* knockdown may affect the expression of genes related to the regulation of the cell cycle and proliferation, which supports the outcome of our in vitro experiments presented above. Moreover, silencing of *CRNDE* was found to influence the regulation of DNA replication and transcription and, in line with the literature data, to alter the expression of genes associated with the cell differentiation and epithelial–mesenchymal transition, as well as with the Wnt/ß-catenin and PI3K/AKT/mTOR pathways. Our study also demonstrated that the silencing of *CRNDE* disturbed the expression of genes encoding proteins that formed the actin and microtubular cytoskeleton and were involved in adhesion. This, once again, supports the results of the above-described microscopic analyses, proving the impact of changed *CRNDE* expression on the shape of cells and their adhesion to a substrate.

### 2.11. Validation of the NGS RNA-Seq Results

The RT-qPCR validation of our NGS results for *CRNDE*, three up-regulated genes (fold change (FC) > 3/2, i.e., *BRCA1*, *HMGA2*, *ZEB1*), and three down-regulated genes (FC < 2/3, i.e., *ALDH1A1*, *CD24*, *CLMN*) was conducted successfully. Firstly, this experiment confirmed the decreased *CRNDE* expression in the SK-OV-3 cells, stably expressing the *CRNDE*-silencing hairpin. Secondly, we managed to identify a strong linear correlation between the expression of all the genes assessed with either NGS or RT-qPCR (Spearman’s correlation test: r = 0.857; *p*-value = 0.0137, Appendix A).

### 2.12. Validation of the CRNDEP Sequence and Identification of Its Interactants by MS

The MS validation of the CRNDEP sequence in one OvCa sample was tricky due to the significant contamination of the IP eluate with antibodies. However, a fractionation of the tryptic peptides prior to the MS analysis, followed by the usage of an inclusion list, led to the identification of 52 CRNDEP amino acids (aa) by matching the measured precursor ions to the masses of six CRNDEP fragments (Appendix A). This covers either 62% or 73% of the entire CRNDEP aa sequence, depending on whether the 84-aa isoform (predicted in our previous study [1]) or the shorter 71-aa isoform, identified recently [6], is considered.

CRNDEP interactants were identified in HeLa cells, for CRNDEP expressed either endogenously or ectopically, by co-immunoprecipitation followed by MS. The joined list of 121 CRNDEP interactants is shown in Appendix A. Thirty-five of these proteins were found to interact with the endogenous CRNDEP variant only, while seventy-two were specific to CRNDEP overexpressed in a fusion with the Strep-Flag tags. However, 14 interactants were found to be partners of endogenous and ectopically overexpressed CRNDEP alike. Interestingly, the CRNDEP micropeptide itself was identified in HeLa cells exclusively when it was overexpressed in a fusion with the tags (Appendix A). This may be due to the fact that endogenously expressed CRNDEP is unstable, as it was demonstrated for the total protein lysate obtained from one high-grade OvCa (hgOvCa) sample (Appendix A). CRNDEP probably becomes even more labile after purification, which hampers its analysis with the use of standard proteomic techniques, e.g., Western blotting [1]. Given that the instability of CRNDEP (predicted by the ProtParam algorithms on the Expasy server as well) might negatively affect the identification of CRNDEP interactants, all our downstream in silico analyses were conducted for the list of proteins co-immunoprecipitated with ectopically overexpressed CRNDEP.

### 2.13. Ontology Analyses for CRNDEP Interactants

Similarly to gene ontology analyses carried out for NGS RNA-seq data, the proteomic data were also evaluated in terms of cellular components, molecular functions, biological processes, and pathways using the EnrichR web application. The ontology analysis results for the proteomic data were, in general, consistent with those for the transcriptomic dataset (Appendix A). In detail, CRNDEP interactants were the proteins forming the cytoskeleton and involved in DNA and RNA metabolism. The cellular components with the highest enrichment in CRNDEP partners were, e.g., stress granules, chaperonin-containing tailless complex (CCT/TriC), focal adhesions, microtubules, chromatin-remodeling complexes (ISWI and NURF), nuclear bodies, and nucleoli. The highly enriched molecular functions included RNA binding and metabolism and also binding of adhesion molecules, like cadherin and vinculin. As to the biological processes involved, the most significantly enriched ones were associated with chromosomes and telomeres functioning, RNA biosynthesis, modification, and degradation in the Cajal bodies, as well as the assembly of stress granules. Finally, the evaluation of the most enriched biological pathways pointed to CCT/TriC-dependent folding of actin and tubulin and the pathways controlling centrosomes’ forming and maturation.

### 2.14. Verification of CRNDEP Interactions In Vitro and In Situ

RNA metabolism was one of the most enriched ontological terms, while the ATP-dependent RNA helicase DDX6 (alternatively named RCK or p54), a protein involved in RNA metabolism, was relatively high on the lists of interactants identified by MS for the endogenous and ectopically overexpressed CRNDEP alike (Appendix A). Considering that p54 is present in stress granules as well, the organelles in which CRNDEP was also found in our previous research [1], we decided to verify interactions between these two proteins in vitro by their co-immunoprecipitation on a magnetic resin coated with streptavidin. In this experiment performed in HeLa cells, both interactants were ectopically expressed, CRNDEP in a fusion with the Strep and Flag tags, and p54 conjugated with RFP (red fluorescent protein). The Western blotting results of detecting RFP in the test and control HeLa cells, presented in Figure 7A, confirmed interactions between CRNDEP and p54. Furthermore, we also managed to corroborate these interactions in situ by conducting co-localization experiments in SK-OV-3 cells with the stable expression of the Flag-CRNDEP fusion protein, and p54-RFP expressed ectopically (Figure 7B,C). A subsequent statistical analysis proved that both proteins, in fact, interacted in the cells, though their co-localization was only partial (Pearson’s correlation coefficient, PCC = 0.278 ± 0.228; Manders overlap coefficients: M1 = 0.639 ± 0.356 and M2 = 0.551 ± 0.181).

RNA metabolism is inseparably associated with nucleoli, and these organelles were among the cellular components mostly enriched in proteins identified in our MS data. This prompted us to evaluate the in situ co-localization between CRNDEP and two markers of the nucleolus, nucleolin, and fibrillarin, both ectopically expressed in HeLa cells. The obtained results demonstrated that treating the cells with actinomycin D (transcription inhibitor) triggered interactions between CRNDEP and both these markers in nucleoli (Figure 8). Noteworthy, for nucleolin, this physiological effect was corroborated by PCC only. Nevertheless, such a confirmation is sufficient for the co-localization to be proven [10].

### 2.15. Assessing RNA Binding Capabilities of CRNDEP In Silico

In our previous research [1], we demonstrated that after ectopic overexpression of CRNDEP in the form of a fusion protein (DsRed Monomer-6xHis-CRNDEP) in HeLa cells, stress granules appeared within the cells and the CRNDEP-containing fusion protein was their component. Apart from the stress granules’ formation, the above-described results of our ontological analyses for both NGS and MS data also portrayed CRNDEP as the micropeptide strongly involved in RNA metabolism. Therefore, an in silico analysis with the use of the SONAR web application was carried out to evaluate the RNA binding capabilities of CRNDEP. The outcome, presented in Appendix A, was consistent with the other CRNDEP-related results shown herein. The statistical model generated in the course of this analysis had good discriminating abilities, allowing to distinguish RNA-binding from non-binding proteins with the sensitivity and specificity of 0.69 and 0.74, respectively (RNA binding protein classification score (RCS) cut-off point = 0.66; AUC = 0.78). The RCS value estimated with the same model for CRNDEP was 0.87. Thus, this micropeptide is likely able to bind RNA molecules.

## 3. Discussion

The aim of this study was to investigate the molecular role of the *CRNDE* gene, its transcripts, and the CRNDEP peptide in selected aspects of carcinogenesis. For this purpose, in A2780 and SK-OV-3 ovarian cancer cell lines, we analyzed the impact of alterations in CRNDE(P) expression on the cell cycle, proliferation, and cell response to selected cytostatics disrupting microtubule metabolism. Moreover, in the SK-OV-3 cells with *CRNDE* knockdown, transcriptome changes were also assessed. In order to understand the physiological role of CRNDEP, protein partners of this micropeptide were searched for as well. Herein, we corroborated the oncogenic role of *CRNDE*, portraying it as the gene impacting carcinogenesis at the stages of DNA transcription and replication, affecting the RNA metabolism, and stimulating the cell cycle progression and proliferation, with CRNDEP being detected in the centrosomes of dividing cells. We also showed CRNDEP to be located in nucleoli and revealed interactions of this micropetide with p54, an RNA helicase. Moreover, we found that CRNDE(P) may affect the interaction of cells with the extracellular matrix, regulate the cell–substrate adhesion process, as well as determine changes in intracellular transport. In addition, ectopic overexpression of CRNDE(P) contributed to an increased rate of microtubules’ repolymerization and higher resistance to cytostatics disrupting their metabolism. Finally, according to our in silico computations, CRNDEP is likely able to bind RNA. All these discoveries perfectly align with the generally acknowledged oncogenic role of *CRNDE* [11].

Our analyses of cell viability carried out with the MTT test, as well as the determination of the mitotic index, showed that the reduction of CRNDE(P) expression in the SK-OV-3 cells significantly inhibited their proliferation rate and metabolic activity. Accordingly, forced expression of CRNDE(P) contributed to the faster rate of cell proliferation and to an increase in the number of dividing cells. Our results align with the outcome of the study by Wang et al. [12], who demonstrated *CRNDE* silencing in the SK-OV-3 cells to be associated with decreased proliferation, invasion, and migration via the miR-423-5p/FSCN1 axis. Other teams showed that increased expression of *CRNDE* stimulates cancer cells to divide, invade other tissues, and metastasize by activation of the Wnt/ß-catenin [13,14], PI3K/AKT/mTOR [15,16], Ras/MAPK [17,18] and Notch1 pathways [19]. The oncogenic role of *CRNDE* depends on the presence of lncRNA transcripts encoded by this gene, which, by interacting with different microRNAs, can inhibit their anticancer activity, acting as specific molecular sponges. Such a regulatory mechanism has been observed in, e.g., breast cancer, where high *CRNDE* expression diminished the level of miR-136, thus activating the Wnt/β-catenin pathway, contributing to the up-regulation of c-Myc and cyclin D1 and resulting in increased rate of cancer cell proliferation [20]. In kidney cancer [21] and hepatocellular carcinoma [22], a mechanism involving the same signaling pathway was also described. As demonstrated in other studies on colorectal cancer, the *CRNDE*-driven activation of the Wnt/ß-catenin pathway lowered the levels of miR-217 [14] and miR181a-5p. Consistently, both *CRNDE* silencing and miR-181a-5p overexpression diminished the proliferative potential of the colorectal cancer cells and sensitized them to chemotherapy [23]. The results of transcriptomic studies presented here indicate that the *CRNDE* gene silencing affected the expression of genes related to the regulation of the Wnt/ß-catenin pathway, including transcripts encoding proteins from the SFRP, DVL, and SMAD families. The expression of genes involved in the Wnt pathway itself was also changed, e.g., the levels of transcripts coding for TCF4, the protein promoting the Wnt signaling, were significantly reduced. In turn, diminished expression of the *MYC* gene upon *CRNDE* knockdown was concordant with the above-mentioned study by Huan et al. [20]. In contrast to the results presented in two studies [20,21], the outcome of transcriptomic analyses presented in this paper showed an increase in the level of cyclins D1 and E1 (as well as A1) elicited by the *CRNDE* knockdown. The observed discrepancy appears to be caused by the fact that the TP53 protein is not expressed in SK-OV-3 cells due to a null (frameshift) mutation in exon 4 of the *TP53* gene (GRCh38:chr17:g.7676106del, ENST00000269305.9:c.267del, ENSP00000269305.4:p.Ser90ProfsTer33, rs587783062). This genetic alteration was found in all control and test clones of the SK-OV-3 cell line used herein. Notably, the same mutation in SK-OV-3 cells was previously reported in the literature [24]. Given the vast network of TP53 protein interactions [25], TP53’s absence, like in SK-OV-3 cells, may seriously alter the cell transcriptome and interactome. In line with this assumption, the expression of *CRNDE* appears to depend on the TP53 status. In our preliminary experiment (Appendix A), we detected the highest and the lowest expression of *CRNDE* in SK-OV-3 (with the null *TP53* mutation) and A2780 (with wild-type *TP53* [26]) cells, respectively. Notably, the difference in *CRNDE* expression between these two cell lines was approximately 80-fold, suggesting that the TP53 protein may control (limit) the level of *CRNDE* transcripts in the cell.

While investigating the G1/S cell cycle checkpoint, other researchers showed that the expression level of the *CCND1* gene in the SK-OV-3 cell line is significantly elevated compared to normal ovarian epithelium [27]. In accordance, in serous hgOvCa, the presence of a frameshift mutation in *TP53*, c.983delT, p.Phe328fs, was associated with the amplification of the *CCND1* gene [28]. Our NGS results seem to align with both these studies, demonstrating an increase in the expression of cyclins A, D, and E in the SK-OV-3 cells after *CRNDE* knockdown. We hypothesize that this phenomenon may be a method to compensate for unfavorable (from the cancer cell’s point of view) physiological changes, resulting in a diminished proliferation rate. Consistently, we observed that *CRNDE* silencing diminished the proliferative potential of the cells, elevating the percentage of cells remaining in the G0/G1 phase and concurrently decreasing the number of cells in the S phase, which was confirmed by other researchers too [29]. Moreover, it has been shown in the literature that *CRNDE* overexpression causes an opposite physiological effect, increasing the subpopulation of cells in the S phase [30]. In this context, our results, demonstrating higher expression of genes related to DNA replication initiation, *E2F1*, *E2F3*, *E2F4*, and *E2F7*, in the cells with *CRNDE* knockdown, may seem intriguing. According to literature data, proteins encoded by the above-mentioned genes perform antithetical functions in the cell, as both E2F1 and E2F3 factors stimulate DNA replication, while E2F4 may act as a repressor in this process [31]. Perhaps this apparent discrepancy is also due to the lack of TP53 protein in the studied cell line. Noteworthy, in mouse brains with *CRNDE* Δ/Δ deletion, altered expression of genes dependent on transcription factors from the E2F family was also demonstrated [32].

The ambiguous impact of *CRNDE* silencing on the cell cycle, found herein, refers not only to the above-discussed G1/S checkpoint but also to the second cell cycle control point, G2/M. On the one hand, the *CRNDE* knockdown in the SK-OV-3 cells elicited overexpression of the *WEE1* gene, encoding a protein kinase that activates this checkpoint [33]. On the other hand, a significantly decreased expression of the *PKMYT1* gene, which encodes another kinase (MYT1) with a similar function [34], was also found in our RNA-Seq data. Two oncogenes promoting progression from G2 to M phase (*CDK7* and *CDC25B*) show a similar mismatch of expression profiles after *CRNDE* silencing. The expression of the former is significantly reduced, which is consistent with the observed decrease in the expression of cyclin H (the CDK7 protein forms a complex with this cyclin, which promotes cell division [35]). However, at the same time, the concentration of the *CDC25B* gene transcript increases significantly, which in turn should exert an opposite, pro-proliferative physiological effect [36]. Similarly to the G1/S checkpoint, the discrepancies in the expression of genes encoding proteins that control the transition from the G2 to M phase of the cell cycle may be caused by the lack of TP53 protein in SK-OV-3 cells, which is known to regulate the activity of both checkpoints [37]. Interestingly, Ding et al. [30] showed in colorectal cancer that the lncRNA encoded by the *CRNDE* gene can epigenetically inhibit the expression of the *CDKN1A* gene, the protein product of which, p21, is a tumor suppressor, interacting with TP53, involved in the control of both aforementioned checkpoints, too. A similar finding was made in the lung adenocarcinoma [38], where it was established that, by interacting with the polycomb repressive complex 2 (PRC2), which maintains overall transcriptional repression in the genome, *CRNDE* lncRNA may down-regulate the *CDKN1A* expression, leading to a low histological differentiation of the tumor, its increased resistance to radiotherapy and tendency for metastasis. The *CRNDE* gene knockdown presented in our study seems to confirm its relationship with PRC2 since, after *CRNDE* silencing, the number of genes with elevated expression was significantly higher than those with reduced expression. In accordance with the above literature reports, our RNA-Seq analysis revealed that one of the overexpressed genes after *CRNDE* knockdown was *CDKN1A*. Considering that we did not detect any mutations in this gene in the SK-OV-3 cell line, one could expect that its high expression would contribute to a higher level of the functional p21 tumor suppressor, which in turn would result in inhibition of the cell cycle, even in the absence of the normal TP53 protein (that under physiological conditions activates the expression of the *CDKN1A* gene [37]). The results of our in vitro studies, shown herein, are fully accordant with this hypothesis.

In our previous paper [1], we showed the high expression of CRNDEP in intensively dividing tissues, such as intestinal crypts, endometrium in the proliferating phase, seminiferous tubules of the human testis, or serous ovarian cancer cells. Here, we discovered a changeable, cell phase-dependent localization of CRNDEP in the cell. In the early G1 phase, the micropeptide is fairly evenly distributed in the nuclear matrix (except for the nucleoli). In turn, in the late G1 and S phase, CRNDEP moves to the nucleoli, whereas in dividing cells, it localizes in the centrosomes, organelles strongly associated with the regulation of cell division and progression of the cell cycle [39]. Such temporal localization changes may suggest that CRNDEP plays a pivotal role in the proliferation process, presumably in the synthesis and maturation of ribosomal RNA in nucleoli. The results of ontological analyses presented in this paper seem to corroborate this assumption, showing that genes encoding proteins related to DNA transcription are significantly overrepresented in the set of genes with altered expression upon *CRNDE* silencing. Our in situ experiments in HeLa cells also confirmed these findings because the addition of actinomycin D (a transcription-blocking antibiotic) to the cell culture resulted in CRNDEP translocation to the nucleolus. Consistently, the identification of CRNDEP micropeptide’s partners supported not only its participation in RNA metabolism but also its centrosomal localization. On the one hand, our co-immunoprecipitation results proved the interaction of CRNDEP with the p54 protein, a helicase involved in the formation of cytoplasmic processing bodies related to the post-transcriptional processing of RNA [40]. This may suggest that the CRNDEP peptide itself also has the ability to bind RNA, which seems to be preliminarily corroborated by our bioinformatic analyses performed with the SONAR program. On the other hand, the list of proteins identified herein by MS as potential CRNDEP partners is significantly enriched in proteins related to the G2/M checkpoint and the functioning and maturation of centrosomes. One such protein, NuMA, is involved in the attachment of the minus ends of the microtubule to the mitotic spindle, which is crucial for its formation and function [41,42]. Another protein possibly interacting with CRNDEP, Plk1, is a serine-threonine kinase, which, among other functions, is responsible for centrosome maturation, mitotic spindle formation, and inactivation of anaphase-promoting complex inhibitors [43]. The third plausible interactant of CRNDEP, the Nlp protein, plays an important role in the formation and elongation of microtubules, centrosome maturation, and the formation of the mitotic spindle. High Nlp expression has been proven to contribute to the development of paclitaxel resistance in breast cancer patients [44].

Ontological analyses of our NGS RNA-Seq data revealed the enrichment in genes encoding both actin and microtubular cytoskeleton proteins, as well as those forming adhesion plates and junctions between the cell and the extracellular matrix. Likewise, among the potential partners of CRNDEP detected by MS, proteins interacting with chaperones from the prefoldin family and with the TriCT/CCT complex, involved in the folding of both tubulin and actin [45], were significantly overrepresented. Consistently, our studies on SK-OV-3 and A2780 cells showed that forced expression of CRNDE(P) could induce resistance of these cells to cytostatics disrupting microtubule dynamics, such as paclitaxel, nocodazole, and noscapine, but not vinblastine. The differences in the sensitivity of the tested cell lines to various cytostatics are probably due to the distinct mechanisms of action of each drug [46]. In the context of the centrosomal location of CRNDEP, described in this paper, the impact of noscapine on the destabilization of the centrosomal γTuRC complex, being responsible for the process of microtubule de novo formation [47], may be particularly interesting. Noteworthy, the disturbance of centrosome functions is a feature specific to noscapine, not described for other cytostatics tested herein [9]. The synergistic effect of combined treatment of the cells with noscapine and paclitaxel/docetaxel demonstrated not only in this paper but also in another study on ovarian cancer [9] may be due to the fact that this type of combination therapy uses two drugs with different target points. This makes cells more sensitive to their effects and, at the same time, makes it difficult for cancer to develop resistance to treatment. It is worth recalling here that in the SK-OV-3 cell line with forced CRNDE(P) overexpression, we observed a clearly increased rate of microtubule repolymerization after removing nocodazole from the culture medium compared to the control line. This result may suggest that the CRNDEP micropeptide accelerates the dynamics of microtubules’ remodeling and/or increases the rate of their de novo formation in centrosomes. This hypothesis would also explain the increased resistance of ovarian cancer cells to most of the cytostatics tested in this work after the *CRNDE* gene overexpression.

Changes in the shape and size of SK-OV-3 cells, observed in this study, caused by differentiation of CRNDE(P) expression, may potentially be associated with disorder within the microtubular cytoskeleton, the activity of which significantly determines cell morphology [48]. The outcome of our transcriptomic analyses showed that *CRNDE* silencing resulted in the down-regulation of 8 out of 20 genes associated with the plus end of the microtubule. This group included, among others, genes related to the regulation of microtubule binding to the mitotic spindle and kinetochore. The list of genes with reduced expression was also significantly enriched in genes encoding proteins involved in the formation of the microtubular and actin cytoskeleton and related to the vesicular transport of various substances within the cell and to the extracellular space. It is worth mentioning here that other teams proved that the transport of many tumor suppressors, such as Rb, BRCA1, or TP53, as well as proteins related to DNA repair, involves the microtubular cytoskeleton [49]. This may explain the greater effectiveness of combined therapy (based on drugs that damage DNA (platinum derivatives) and disrupt cell division (taxanes), routinely used in the treatment of ovarian cancer) than monotherapy [50].

Disturbance of interactions between *CRNDE*-silenced cells and the extracellular matrix is evidenced in our study by a decrease in the number and the total size of focal adhesion plaques, determined based on the immunodetection of paxillin. However, transcriptomic analyses performed for the same cell line did not reveal changes in the expression of *PXN*, the gene that encodes paxillin. One explanation for this phenomenon may be a decrease in the stability of paxillin due to the *CRNDE* knockdown. The observed aberrations in the substrate adherence of the cells with reduced CRNDE(P) expression may also be caused by incorrect delivery and anchoring of paxillin to the cell membrane, which could be partly due to incorrect vesicular transport. An in-depth ontological analysis of our NGS RNA-seq results presented in this paper proves that a low *CRNDE* level probably contributes to the impairment of cell–substrate adhesion, resulting from a decreased expression of genes encoding proteins involved in the formation and maturation of adhesion plaques, including talin, filamin, and tensin. These proteins participate in the binding of actin filaments to the extracellular matrix via integrins [51]. Small GTPases from the Ras superfamily of proteins are also involved in the formation of adhesion plaques, including the RhoA protein [52], encoded by the *RHOA* gene, the transcript level of which was also significantly reduced in the *CRNDE*-silenced SK-OV-3 cells. Moreover, the aforementioned talin is a mechanosensor responsible for the maturation of adhesion plaques and the increase in their area [53]. The ontological analysis of the proteins identified as potential CRNDEP partners was consistent with the results of the transcriptomic analyses described above. It confirmed the relationship between the tested micropeptide and the metabolism of adhesion plaques, showing a statistically significant enrichment in proteins involved in their formation and maturation, such as 14-3-3E, 14-3-3G, PDIA3, heterogeneous nuclear ribonucleoprotein K, cofilin, α-catenin, ezrin, or PDLIM7. One of these proteins, ezrin, mediates the formation of a bond between the cell membrane and the actin cytoskeleton and plays a key role in cell migration [54]. Another potential partner of CRNDEP, cofilin, is responsible for the polymerization and depolymerization of actin [55]. It also determines the correct central arrangement of the mitotic spindle and plays a role in the regulation of the morphology and organization of the cytoskeleton in epithelial cells [56]. Like ezrin, cofilin is overexpressed in numerous malignancies, including breast and ovarian cancers [57]. The results of cell adhesion studies reported by other researchers seem to confirm our observations, proving that multiple myeloma cells lacking the *CRNDE* gene (CRNDEΔ/Δ) are characterized by a significantly reduced ability to adhere to the substrate compared to myeloma cells expressing this gene at an unchanged level. Moreover, transcriptomic analyses carried out by the authors of the discussed paper also showed that the lack of *CRNDE* in the cells led to a significant reduction in the expression of genes related to cell–cell and cell–extracellular matrix interactions [58].

CRNDEP belongs to the class of micropeptides known as SEPs. Unfortunately, despite their indisputable physiological importance [7], unambiguous identification of SEPs is difficult due to the low frequency of smORF sequences in currently available databases. The underestimation of their number results from the common use of quite rigorous criteria for the inclusion of transcripts in such databases, which is highly suboptimal for smORFs identification [59]. The use of new approaches for analyzing smORFs, employing comparative genomics [60], new proteomic techniques such as peptidomics, and the combination of bioinformatic data on evolutionary-conserved ORF sequences with the results of ribosome profiling (Ribo-Seq) [3], revealed that the number of smORFs that can be translated probably far exceeds previous predictions [61]. Consistently, the criteria for MS analysis and subsequent inclusion of sequences in proteomic databases should also be changed because the standard rules used today eliminate micropeptides automatically due to the small number of identified fragmentation spectra [62]. Thus, the inability to clearly identify a specific micropeptide with currently available methods should not be considered proof of its non-existence.

Notably, out of 102 human Ribo-Seq studies listed in the RPFdb database, only 10 were carried out on primary biological material and not on cell lines, which can also hamper the discovery of novel micropeptides [6]. To overcome this limitation, Chothani et al. [6] have generated an ultra-high-depth RNA- and Ribo-Seq dataset across six human primary cell types and five human tissues and developed a pipeline to identify smORFs. This was further combined with the evaluation of the evolutionary conservation of found aa sequences and their MS-based validation. This approach confirmed the presence of over 600 novel micropeptides in human cells, including CRNDEP. Remarkably, the authors of the cited study [6] managed to identify in their Ribo-Seq data only 71 of 84 aa residues theoretically forming CRNDEP and then confirmed the existence of a 24-aa oligopeptide located in the central region of this micropeptide by MS/MS. Herein, we not only confirmed the results obtained in the aforementioned study, but we also managed to extend the MS-detected region to 52 aa. Though, it needs to be stressed that our MS identification of CRNDEP was based on the analysis of precursor ions only since the quality of mass spectra was too low (likely due to the instability of this micropeptide) to successfully perform a complete MS/MS analysis. Noteworthy, the 30-aa N-terminal region of CRNDEP has not been confirmed yet by MS in any study. However, the anti-CRNDEP antibody utilized in our research should be capable of binding both the 84-aa and 71-aa isoforms of CRNDEP with the same specificity since the synthetic 15-aa epitope used for the antibody development matches the central region of CRNDEP. This region starts from Asp29 and Asp16 in the 84-aa and 71-aa isoforms, respectively.

## 4. Materials and Methods

### 4.1. Cell Lines

In the present study, four human ovarian cancer cell lines, SK-OV-3, A2780, IGROV1, and TOV-112D, were used. They were derived from serous (SK-OV-3) and endometrioid (all the remaining cell lines) ovarian carcinomas [63]. The first two cell lines listed were particularly interesting as subjects of a profound molecular analysis. The SK-OV-3 cells are the model of serous ovarian cancer, being the most frequently occurring ovarian malignancy in humans. Moreover, this cell line harbors a homozygous frameshift mutation in the *TP53* gene, leading to the complete loss of the TP53 tumor suppressor protein (null mutation). By contrast, A2780 is the only ovarian cancer cell line assessed herein, which has the wild-type *TP53* gene (the IGROV1 and TOV-112D cells harbor missense mutations in *TP53*, while IGROV1 cells additionally have a single-nucleotide duplication in the coding region of this gene) [63]. Given their molecular characteristics and the fact that SK-OV-3 and A2780 cells exhibited the highest and the lowest *CRNDE* gene expression levels, respectively (Appendix A), these two cell lines were selected as cellular models in the present study. Daughter cell lines with stable silencing of *CRNDE* were derived from the SK-OV-3 cell line only, while daughter cell lines with *CRNDE* overexpression were developed from both the SK-OV-3 and A2780 cells. For each daughter cell line, a corresponding control cell line was also established. Aside from the aforementioned ovarian cancer cell lines, the HeLa cell line, originating from a human cervical cancer, was employed in some experiments, as well. All the parental cell lines have been successfully authenticated by DNA short tandem repeat (STR) profiling. For details, see Section 4.21.

### 4.2. Establishment of Cell Lines Expressing the CRNDEP Micropeptide Fused to the Double Flag Tag

A construct expressing the CRNDEP micropeptide in a fusion with the double Flag tag (Flagx2) was introduced into the AAVS1 safe harbor locus of the SK-OV-3 and A2780 ovarian cancer cell lines using the CRISPR/Cas9 technology. To facilitate the recombination of the transgene into the genome, the AIO-GFP backbone plasmid (Addgene, Watertown, MA, USA; cat. no. 74119) was used. This plasmid encodes dual U6 promoter-driven sgRNAs and also EGFP-coupled Cas9-D10A endonuclease, mutated in such a way as to generate just a single nick in the target DNA molecule, thus decreasing off-target effects. The sgRNA-coding DNA insert (sgRNA T2) with sticky ends allowing for its ligation with the AIO-GFP plasmid was obtained from two mutually complementary oligonucleotides (sgRNA_AAVS1_TOP, sgRNA_AAVS1_BOT) purchased at oligo.ibb.waw.pl (Warsaw, Poland). For their sequences, and also for sequences of other synthetic DNA molecules utilized herein, refer to Appendix A. Aside from the insertion of the sgRNA T2 region into the AIO-GFP backbone plasmid, the *EGFP* gene was cut out using the EcoRI and NotI restriction enzymes, followed by filling in the sticky 5′ ends with the Klenow fragment of the DNA polymerase I from *Escherichia coli*, and subsequent circularization of the resultant molecule by ligation to form the AIO-sgRNA-AAVS1 plasmid (Appendix A). The SK-OV-3 and A2780 cells were subsequently co-transfected (applying the Lipofectamine 2000 transfectant (Thermo Fisher Scientific, abbreviated as Thermo, Waltham, MA, USA)) with 1 µg of the AIO-sgRNA-AAVS1 vector and 3 µg of one of the following transgene-harboring plasmids: AAVS1-CAG-hrGFP, AAVS1-CAG-FLAGx2, and AAVS1-CAG-FLAGx2-CRNDEP (Appendix A). The AAVS1-CAG-hrGFP plasmid was used to evaluate the efficacy of the CRISPR/Cas9-dependent recombination of the *hrGFP* gene into the AAVS1 locus within the HeLa cells’ genome (Appendix A). The remaining two transgene-harboring constructs were utilized to create either the control cell lines expressing the double Flag tag or the cell lines with the Flagx2 tag fused to the N-terminus of the CRNDEP micropeptide, respectively. Twenty-four hours after transfection, the RPMI-1640 culture medium (BioWest, Lakewood Ranch, FL, USA) was supplemented with puromycin in a final concentration of 2 μg/μL (the A2780 cells) or 6 μg/μL (SK-OV-3 cells), and the RS-1 factor (final conc. 7.5 nM), the RAD51 activator, stimulating the homology-directed repair (HDR) of double-strand DNA lesions. The puromycin concentrations were earlier determined experimentally. The clonal selection of recombinants was performed in 96-well culture plates 72 h after puromycin was added to the medium. Finally, the expression of the transgenes incorporated into the human genome was assessed by either Western blotting or dot blotting, using the primary anti-Flag antibody conjugated with horseradish peroxidase (HRP) (Sigma-Aldrich, St. Louis, MO, USA) and the custom-made primary rabbit polyclonal IgG CRNDEP-specific antibody, developed for us by Abgent, Inc. (San Diego, CA, USA) [1], subsequently detected with the donkey anti-rabbit HRP-conjugated secondary antibody (GE Healthcare, Chicago, IL, USA). For detailed information on antibodies utilized in the present study, including their concentrations, see Appendix A. The enzymatic activity of HRP was visualized with the WesternBright Quantum HRP substrate (Advansta, Menlo Park, CA, USA) on the UVITEC Essential D77 Documentation System (UVITEC, Cambridge, UK).

### 4.3. Development of SK-OV-3 Cell Lines with the CRNDE Gene Knockdown

The RNA interference-based down-regulation of the *CRNDE* gene expression in the SK-OV-3 cell line was obtained by using the pGFP-B-RS backbone vector (OriGene Technologies, Inc., Rockville, MD, USA), allowing for the incorporation of DNA inserts encoding shRNA molecules, subsequently being expressed from the U6 promoter. In the present study, four shRNA-coding dsDNA inserts were generated by denaturation and subsequent renaturation of synthetic mutually complementary oligonucleotides obtained from oligo.ibb.waw.pl. Three of them (SH1, SH2, and SH3) encoded shRNAs suitable to silence the expression of the *CRNDE* gene, while the last one (SH SCR) was used as a negative control (Appendix A). Each shRNA was designed by our team with the siRNA Wizard v3.1 web application (http://www.invivogen.com/sirnawizard/, accessed on 13 March 2014) to diminish the risk of potential off-target effects. All the shRNA-coding inserts had sticky ends specific to digestion with either the BamHI or HindIII restriction enzyme. This enabled their ligation in the correct orientation with the pGFP-B-RS backbone vector, previously cleaved with the same set of enzymes. Each of the resultant constructs harbored the blasticidin resistance gene and allowed for the expression of not only the appropriate shRNA molecule but also of the gene encoding the TurboGFP reporter protein. The correctness of inserts’ incorporation into the pGFP-B-RS vector was verified by Sanger sequencing using two different primers (promoU6_F, SV40_R, Appendix A) and the BigDye Terminator v.3.1 Cycle Sequencing Kit (Thermo) in conditions recommended by the kit’s manufacturer. Due to a high risk of secondary structures’ formation within the sequenced templates, the standard sequencing buffer was supplemented with DMSO (final concentration: 5%) and dGTP (final concentration: 40 µM). Before the analysis on the ABI PRISM 3130xl Genetic Analyzer in the POP-7 polymer (both manufactured by Thermo), the sequencing products were cleaned with the use of the ExTerminator kit (A&A Biotechnology, Gdansk, Poland).

Each of the above-mentioned silencing/control constructs was introduced into the SK-OV-3 cells by transfection with Lipofectamine 2000. The presence of 5′ and 3′ LTR regions from the MMLV (Moloney Murine Leukemia Virus) encompassing each construct enabled its efficient, retrotransposition-based incorporation into the cell line’s genome. Forty-eight hours after transfection, a clonal selection of recombinants was made in the selective RPMI-1640 medium, supplemented with 10% (*v*/*v*) fetal bovine serum (FBS, BioWest), 1% (*v*/*v*) of a ready-to-use penicillin and streptomycin solution (BioWest), and blasticidin in the final concentration of 6 μg/mL (Thermo). The efficiency of CRNDE knockdown in daughter cells was assessed by RT-qPCR on the 7500 Fast Real-Time PCR System (Thermo), using 15 ng of total RNA extracted from each cell line and reverse transcribed to cDNA with the High-Capacity cDNA Reverse Transcription Kit (Thermo). The RT-qPCR experiments were run in triplicates, using the QUANTUM Probe + ROX PCR Kit (Syngen Biotech, Wroclaw, Poland) along with our two in-house designed CRNDE-specific TaqMan assays [2], which are capable of detecting either the CRNDEP-coding transcripts only (i.e., FJ466686.1 and NR_170995.1) or other CRNDEP-non-coding splice variants (FJ466685.1, NR_034105.4 and NR_034106.3). Additionally, commercially available TaqMan assays for two reference genes, HGPRT (cat. no. 4326321E, Thermo) and PPIA (cat. no. 4333763F, Thermo), were used. Conditions of the RT-qPCR reaction were consistent with Thermo recommendations. The ΔΔCt method was applied for the relative quantification of gene expression.

### 4.4. Mitotic Index Evaluation

The mitotic index of the cells was assessed by immunofluorescence staining with the use of an antibody specific to the phosphorylated form of the histone H3 (Ser10), which is characteristic of the dividing cells (such phosphorylation persists between the late G2 phase and anaphase). This parameter was found to outperform Ki67 as a biomarker of breast cancer prognosis [64]. The mitotic cells were counted under the LSM8 confocal microscope with the Airyscan detector (Zeiss, Oberkochen, Germany) with a 40× objective in 5–8 areas, comprising four fields of view each. The number of DAPI-stained nuclei in a single area ranged from 600 to 800. Thus, the total number of cells analyzed for each cell line exceeded 3000. Subsequently, the Kruskal–Wallis test with the Benjamini, Krieger, and Yekutieli correction was applied for statistical inference.

### 4.5. Cell Viability Assessment by the MTT Assay

Cells in the exponential growth phase were treated with the 0.05% trypsin-EDTA solution at 37 °C for 5 min to obtain a cell suspension. Next, the cells were counted in the Neubauer chamber and diluted with PBS before seeding them into a culture plate containing the RPMI-1640 medium to obtain about 80% confluency on the day when the assay was to be carried out. The MTT compound (Sigma-Aldrich) was added to the cells or the wells filled with the medium only (a negative control) in the final conc. of 0.5 mg/mL. Afterward, the cells were grown in a cell incubator (37 °C, 5% CO_2_) for 3 h, followed by the medium removal and addition of 150 µL of DMSO to each well to dissolve formazan crystals by incubating the culture plate at room temperature with shaking for 1 h. Next, the absorbance at 540 nm was measured on the Victor 3 spectrophotometer (model: 1420-012, Perkin Elmer, Waltham, MA, USA). The negative control wells were used as blank samples in this experiment. Finally, the one-way ANOVA test with Dunn’s correction for multiple comparisons was applied.

### 4.6. Cell Cycle Phase-Dependent Changes in CRNDEP Localization

The cell cycle of HeLa cells was stopped in different phases by culturing the cells with 1% FBS for 18 h (G0/G1 block), 400 µM L-mimosine for 24 h (G1/S block), 1 mM thymidine for 18 h (S phase block), or 100 ng/mL nocodazole for 24 h (M phase block).

### 4.7. Evaluation of Cell Distribution between Different Cell Cycle Phases

To determine the frequency of cells remaining in particular cell cycle phases, the flow cytometry analysis of the cells stained with propidium iodide was performed on the FACS AriaIII flow cytometer (Becton Dickinson Biosciences, Franklin Lakes, NJ, USA), in step with the protocol published by Lee et al. [65]. Next, the data were analyzed in the FACSDiva Software v2.5.1 to identify cells being in one of three main phases, i.e., G0/G1, S, and G2/M. Finally, the Kruskal–Wallis test was employed to find out whether the observed differences in cell distribution between the phases were statistically significant.

### 4.8. The Analysis of Cell Shape Changes

Images (comprising 200–300 cells per cell line variant) were captured in transmitted light under the LSM8 confocal microscope using the 40× objective. Next, the images were analyzed in the ImageJ application (v. 1.52r) by manually outlining the cells and then determining their perimeters and areas. This information was then used to calculate the CSI according to the following formula: 4π × cell area [μm^2^]/cell perimeter [μm]^2^. The CSI coefficient equals 1 for perfectly circular cells and tends to 0 for elongated cells. Obtained data have been supplemented with plots portraying the CSI distribution as well as with the results of the Kruskal–Wallis test with a correction for multiple comparisons.

### 4.9. Assessing the Speed of Microtubules’ Repolymerization

Twenty-four hours after seeding, the SK-OV-3 cells were grown in a cell incubator (37 °C, 5% CO_2_) in the RPMI-1640 medium supplemented with 20 μM nocodazole for four h. To make repolimerization of microtubules possible, the nocodazole-containing medium was removed. Then, the cells were placed in the standard medium and grown in a cell incubator for 10 min. Afterward, the cells (including their cytoskeleton) were fixed with methanol at −20 °C for 20 min, followed by their immunofluorescence staining with the alpha tubulin-specific primary mouse antibody (Thermo) and the secondary Alexa Fluor 647-conjugated goat anti-mouse antibody (Thermo), according to the protocol described by Grzybowska et al. [66]. Visualization of the cells was performed under the LSM8 microscope with the 63× objective.

### 4.10. Investigating the Changes in Cell–Substrate Adhesion

In this experiment, the number and the size of focal adhesion plaques (FAs) were assessed by immunodetection of paxillin, which is a protein specific to this structure. FAs were visualized under the LSM8 confocal microscope (with the 40× objective) for 500–1000 cells of each cell line and then counted in ImageJ. The staining procedure and the calculation methods were provided in our previous paper [67]. The statistical inference involved the usage of the one-way ANOVA test with Dunn’s correction for multiple comparisons, which was preceded by the evaluation of the distribution’s normality with the Shapiro–Wilk test.

### 4.11. Cytotoxicity Tests

Twenty-four hours after seeding the analyzed cell lines on a 96-well plate (20,000 cells per well), the cells were treated with one of four different cytostatics, i.e., nocodazole, vinblastine, paclitaxel, and noscapine. In addition, the combination of paclitaxel and noscapine was also used. The concentrations of each compound were selected based on data available in the scientific literature and equaled for nocodazole [nM]: 0, 100, 300, 600; for vinblastine [nM]: 0, 1, 5, 10; for noscapine [µM]: 0, 6.25, 12.5, 25, 50, 100; for paclitaxel [nM]: 0, 6.25, 12.5, 25, 50, 100 (with or without 20 µM noscapine). Cytotoxicity of the drugs was assessed using the MTT assay to calculate the IC50 value for every compound. The MTT test was performed in at least 3 technical and biological repeats 48 h after the addition of the cytostatics to the culture media. The obtained data were then analyzed statistically by applying the two-sided t-Student test for paired samples with the Welch correction for multiple comparisons.

### 4.12. NGS Analysis of Transcriptomic Alterations

For two independently developed SK-OV-3 cell lines with stable silencing of *CRNDE* with the SH1 shRNA (SK-OV-3-SH1-K1, SK-OV-3-SH1-K2) and two corresponding control cell lines (SK-OV-3-SCR-K1, SK-OV-3-SCR-K3), a whole transcriptome evaluation was carried out by NGS RNA-seq in three biological replicates, giving 12 samples in total. This study involved total RNA extraction (from 1 million cells per sample) with the Pure Link RNA mini kit (Thermo). Then, RNA was treated with the Turbo DNA-free Kit (Thermo) to remove all traces of DNA from the samples. RNA concentrations were assessed on the Quantus Fluorometer, using the QuantiFluor RNA System kit (Promega, Madison, WI, USA). RNA quality evaluation by RNA integrity number (RIN) estimation (all the RINs ≥ 9) was carried out with the Agilent RNA 6000 Nano Kit on the 2100 Bioanalyzer (both manufactured by Agilent Technologies, Santa Clara, CA, USA). Next, 1 µg of total RNA was reverse transcribed to cDNA with the SuperScript II kit (Thermo). The obtained cDNA was then used to generate NGS libraries with the TruSeq Stranded Total RNA Library Prep Gold (Illumina, San Diego, CA, USA), according to the recommendations of the kit’s manufacturer. The VAHTS RNA Clean Beads (Vazyme Biotech, Nanjing Shi, China) were applied to clean cDNA after every enzymatic reaction. Subsequently, the distribution of molecules’ lengths in the libraries, as well as the cDNA concentration in each library, were assessed on the 2100 Bioanalyzer by using the Agilent High Sensitivity DNA Kit (Agilent Technologies). Finally, all the libraries were pooled in equimolar ratios to create the multi-library, additionally supplemented with the PhiX Control v3 Library (Illumina) in the final conc. of 1%, and then sequenced on the Illumina NovaSeq 6000 platform in the paired-end mode (2 × 100 bp). The resultant NGS RNA-Seq FASTQ files have been submitted to the European Nucleotide Archive (ENA) database (data acc. no. PRJEB61091).

### 4.13. Bioinformatic Analyses of the NGS Data

First, the obtained FASTQ files had the number of read pairs (median = 67.0 × 10^6^) and the quality assessed with the Fastqc app (v. 0.11.9), followed by the removal of adapters and poor-quality regions with Trimmomatic (v. 0.39). Next, using two different aligners, HISAT2 (v. 2.2.1) and STAR (v. 2.7.9a), the reads were mapped to the reference human genome (v. GRCh38) with over 97% efficiency (median) regardless of the aligner used, giving over 800 reads (median) per gene. Afterward, PCR and optical duplicates were removed with MarkDuplicates (a part of the Genome Analysis Toolkit, GATK, v. 4.1.7.0), and the mapping quality was evaluated with the Samtools (v. 1.12) and Qualimap (v. 2.2.2-dev) apps. Later, in the R environment, heatmaps were generated (package: pheatmap v. 1.0.12), and the principal component analysis (PCA, package: DESeq2, v. 1.26.0) was performed for the log2-transformed gene expression data to check how the expression differed between the analyzed samples. Then, the differential expression analysis between both groups was carried out with two alternative, independent R packages, DESeq2 and edgeR (v. 3.28.1), using the R script (gene_expression.r, v. 1) available for download at https://github.com/lukszafron (accessed on 17 May 2023) (LMS_gh). The employment of two different read aligners and two distinct methods for the assessment of gene expression changes, in combination with the usage of the Benjamini–Hochberg correction (FDR) for multiple comparisons, decreased the risk of type I statistical error. The resultant common gene set (5934 genes) with the expression significantly altered between the SK-OV-3 cells with and without the *CRNDE* knockdown was subsequently utilized in gene ontology (GO) analyses to find the most enriched GO-terms in three categories: biological processes, molecular functions, and cellular components. For this purpose, the ShinyGO v0.741 (http://bioinformatics.sdstate.edu/go/, accessed on 15 December 2021) and EnrichR (https://maayanlab.cloud/Enrichr/, accessed on 15 December 2021) web applications were applied.

To identify genetic variants in the selected genes, e.g., *TP53* and *CDKN1A*, in our RNA-Seq data for SK-OV-3 cells, a previously described multi-tool bioinformatics pipeline [68], utilizing the Ensembl Variant Effect Predictor app (v. 100), was applied. Finally, to perform further bioinformatic analyses, statistical inference, and outcome visualization, two R scripts, vep.r (v. 2.1) and vep.comparison.r (v. 2.1), were used. Both these programs, developed by LMS, are downloadable from LMS_gh.

### 4.14. Validation of the NGS RNA-Seq Results

To check the reliability of our NGS results, the expression of *CRNDE* (in-house designed TaqMan assays [2]), and six other genes with significantly altered expression after *CRNDE* silencing was assessed with RT-qPCR, i.e., *BRCA1* (TaqMan assay (TMA) cat. no. Hs01556193_m1, Thermo), *HMGA2* (TMA cat. no. Hs00171569_m1, Thermo), *ZEB1* (TMA cat. no. Hs00232783_m1, Thermo), *ALDH1A1* (TMA cat. no. Hs00946916_m1, Thermo), *CD24* (TMA cat. no. Hs04405695_m1, Thermo), and *CLMN* (TMA cat. no. Hs00226865_m1, Thermo). The *HGPRT* and *PPIA* housekeeping genes were applied as references for expression normalization. Noteworthy, neither of these two reference genes was found differentially expressed upon *CRNDE* knockdown in the NGS analysis performed herein.

### 4.15. Creation of the pDEST/C-Strep-FlagTags-CRNDEP Plasmid

This plasmid was developed by applying the Gateway recombination cloning technology (Thermo), which is suitable for quick DNA insert cloning. To obtain the entry clone for the Gateway system, the ORF of the CRNDEP peptide was PCR-amplified using the B and L primers (Appendix A) that provided the attB sites encompassing the given ORF and thus allowed for the BP Clonase-catalyzed recombination of this ORF into the pDONR vector. The selection of recombinants was performed by transforming products of the recombination into chemo-competent E. coli DH5α cells, then cultured in the LB Agar medium (Sigma-Aldrich), supplemented with gentamicin (20 μg/mL, Polpharma S.A., Starogard Gdanski, Poland). The final construct was generated in the second round of Gateway cloning (catalyzed by the LR Clonase) when the CRNDEP ORF present in the entry clone was subcloned to the pDEST/C-Strep-FlagTags-TAP vector (a generous gift from Prof. Frank Gaunitz) to form the final pDEST/C-Strep-FlagTags-CRNDEP plasmid (Appendix A). The selection of recombinants was performed by DH5α cells’ transformation followed by their culturing in the LB Agar medium, supplemented with kanamycin (30 μg/mL, Sigma-Aldrich). Both insert cloning rounds were carried out under the conditions recommended by Thermo. The pDEST/C-Strep-FlagTags-CRNDEP plasmid was later employed for identification of CRNDEP protein partners by co-immunoprecipitation coupled with MS.

### 4.16. Identification of the CRNDEP Micropeptide by Immunoprecipitation (IP) Followed by MS

A detailed description of this workflow is provided on the second page of the Appendix A. The mass spectrometry proteomics data have been deposited to the ProteomeXchange Consortium (http://proteomecentral.proteomexchange.org, accessed on 28 July 2023) via the PRIDE partner repository [69] with the dataset identifier PXD044196 and DOI 10.6019/PXD044196.

### 4.17. Identification of CRNDEP Interactants

The CRNDEP micropeptide’s interactome was investigated by co-immunoprecipitation (co-IP) of a total protein lysate from approximately 5 million HeLa cells (obtained by using Nonidet P40 (NP40) buffer (B.D.H. Chemicals Ltd., Poole, Dorset, UK), supplemented with the Halt Protease Inhibitor Cocktail, (Thermo)). The co-IP procedure was carried out in biological triplicates using superparamagnetic beads coupled with the recombinant Protein A (Dynabeads Protein A, Thermo), coated with 10 µg of the anti-CRNDEP antibody, and the Dynal Magnetic Particle Concentrator (MPC-S, Thermo), in compliance with the protocol provided by Balcerak et al. [67]. Noteworthy, the protocol was modified by crosslinking the antibody to the beads with the BS3 compound (Thermo) according to Thermo recommendations. In the control experiment, the anti-CRNDEP antibody was replaced with 10 µg of a rabbit IgG isotype control antibody (Sigma-Aldrich). The co-IP efficiency was assessed by Western blotting. The test and control co-IP samples were subsequently analyzed by MS, as described previously [67], on the QExactive hybrid quadrupole orbitrap mass spectrometer (Thermo) coupled with a nanoAcquity LC system (Waters Corporation, Milford, MA, USA). The mass spectrometry proteomics data have been deposited to the ProteomeXchange Consortium (http://proteomecentral.proteomexchange.org, accessed on 28 July 2023) via the PRIDE partner repository [69] with the dataset identifier PXD044160 and DOI 10.6019/PXD044160. Subsequent bioinformatic analyses were carried out as described in our previous paper [67]. This experiment was performed in two variants. In the first one, HeLa cells expressing endogenous CRNDEP (with no transgene introduced into the cells) were utilized. The second variant involved the cells 24 h after transfection with the pDEST/C-Strep-FlagTags-CRNDEP plasmid, exhibiting an ectopic expression of CRNDEP fused to two tags, Strep and Flag. The list of interactants identified for the ectopically expressed CRNDEP was later used in the ontology analyses performed in the ShinyGO and EnrichR apps.

### 4.18. Confirmation of Interactions between CRNDEP and the p54 Protein

Interactions between CRNDEP and p54 were verified by either co-IP (in HeLa cells) or in co-localization studies (in SK-OV-3-Flag-CRNDEP-K2 cells; see below). The interacting partners, CRNDEP and p54, were expressed from the pDEST/C-Strep-FlagTags-CRNDEP and pCMV-SPORT6-p54-RFP plasmids, respectively. A total of 10 µg of each plasmid was used for a double transfection of 4.5 million HeLa cells with Turbofect (Thermo). Next, the cells were grown in the DMEM medium supplemented with 10% (*v*/*v*) FBS and 1% (*v*/*v*) of the ready-to-use penicillin and streptomycin solution for 24–48 h. This allowed for ectopic expression of both CRNDEP (with the Strep and Flag tags fused to its C-terminus) and p54 conjugated with RFP. In this affinity purification (AP) experiment, a special magnetic resin, MagStrep “type3” XT beads (IBA Lifesciences, Göttingen, Germany), coated with the streptavidin variant Strep-Tactin^®^XT was employed, which made it capable of binding the Strep-tag with a high affinity. The co-IP reaction was carried out according to the recommendations of the resin’s manufacturer. To obtain a negative control for this experiment, the pDEST/C-Strep-FlagTags-CRNDEP plasmid was replaced with the pCR3-2xFLAG-CRNDEP plasmid (Appendix A), which encoded the CRNDEP micropeptide linked to two Flag tags and no Strep-tag. Thus, the control variant of the CRNDEP fusion protein was unable to bind to the aforementioned resin. Other conditions of the cell transfection and co-IP reaction remained unchanged. Ultimately, to confirm interactions between CRNDEP and p54, the co-IP-derived samples were analyzed by Western blotting with the use of the anti-RFP rat monoclonal primary antibody and the goat anti-rat HRP-conjugated secondary antibody (Appendix A).

### 4.19. Co-Localization Studies

In situ co-localization experiments were performed in cell lines grown on sterile, circular 15 mm microscope cover glasses (100,000 cells per glass) (NEST Biotechnology, Wuxi, China) placed in 24-well culture plates. Twenty-four hours after cell passaging or transfection (depending on whether the experiment protocol included this step), the cells were rinsed twice with PBS and then fixed by incubation with 4% formaldehyde for 15 min on ice. The alternative fixation protocol (applied when a cellular cytoskeleton was to be examined) relied on a 20-min incubation of the cells with methanol at −20 °C. Further steps of cell staining were carried out according to the method described in our previous paper [66]. All cell images presented in this paper were captured under the LSM8 confocal microscope. Later, to obtain a numerical representation of the signals’ co-localization degree, the ImageJ software with the JACoP plugin (v. 2.1.1) was employed. This plugin provides the functionality to automatically calculate Pearson’s correlation and Manders’ overlap coefficients for the analyzed signals.

CRNDEP in situ localization was identified with our custom-made anti-CRNDEP antibody, which was subsequently detected with one of two goat anti-rabbit secondary antibodies coupled with either the Alexa Fluor 488 (green) or Alexa Fluor 647 (red) fluorescent dye (both secondary antibodies are offered by Thermo).

Aside from co-IP, the interactions between CRNDEP and p54 were also evaluated by co-localization studies in the SK-OV-3-Flag-CRNDEP-K2 cell line with the stable expression of CRNDEP fused to the Flag tag. The cells were additionally transfected with the pCMV-SPORT6-p54-RFP plasmid (a generous gift from Prof. Dominique Weil), which allows for ectopic expression of the p54-RFP fusion protein.

The interactions between CRNDEP and two nucleoli markers, nucleolin and fibrillarin, were investigated in HeLa cells, first cultured for 4 h in the DMEM medium supplemented with 1 μg/mL DMSO-dissolved actinomycin D (Sigma-Aldrich; test group) or the corresponding amount of DMSO (control group), and then transfected with either the pEGFP-Nucleolin or pEGFP-Fibrillarin plasmid. Both these constructs, generously provided to us by Prof. Roman Szczesny, enabled the expression of the nucleoli markers in a fusion with EGFP. The co-localization of CRNDEP with either marker was assessed for 50 nucleoli (about 20 cells) from each group, and the results were subjected to a statistical analysis with the two-sided t-Student test for paired samples with the Welch correction for multiple comparisons.

### 4.20. Assessing RNA Binding Capabilities of CRNDEP In Silico

To investigate the RNA binding capabilities of the CRNDEP micropeptide, the SONAR web app was utilized. This procedure required the assessment of the RCS for both RNA-binding and RNA-non-binding proteins based on the human interactome data gathered by Hein et al. [70] for HeLa cells. In order to make those data more complete, that interactome was further extended by the proteins identified in our MS analyses as CRNDEP interactants. This approach allowed us to generate a statistical model capable of discriminating the RNA-binding and non-binding proteins with good sensitivity and specificity (ROC AUC = 0.78). The obtained model was then used to determine whether the CRNDEP micropeptide can bind RNA molecules.

### 4.21. Cell Lines Authentication

All parental cell lines employed in the present study (A2780, SK-OV-3, IGROV1, TOV-112D, and HeLa) have been authenticated by DNA STR profiling. The profiling was carried out with the PowerPlex^®^Fusion 6C System kit (Promega) on the GeneAmp PCR System 9700 (Thermo). Then, PCR amplicons were separated in the POP-4 polymer on the ABI Prism 3500xl DNA sequencer and analyzed with the GeneMapper ^®^ ID-X v1.4 app (the sequencer, reagents, and software were all provided by Thermo). At all stages of the DNA profiling procedure, both positive and negative controls were used. Each cell line was considered positively authenticated if the identified alleles were consistent with the cell line-specific STR profiles deposited in the Cellosaurus.org database [63] for at least 13 [71] out of 24 loci analyzed herein. For the results, refer to Appendix A.

## 5. Conclusions

In this research, by performing the MS analysis, we detected the CRNDEP micropeptide, the existence of which was uncertain until recently. Moreover, our results expand the current state of knowledge on the physiological functions of the *CRNDE* gene expression products and their role in tumorigenesis. Particularly interesting are the results showing the relationship between high CRNDE(P) expression and increased resistance of ovarian cancer cells to treatment with cytostatics that disrupt microtubule dynamics. Moreover, this work provides important new information on the stimulating effect of *CRNDE* gene expression products on the cell proliferation rate, the activity of the microtubular cytoskeleton, and the ability of the cells to form adhesion plaques. Finally, according to our in silico analyses, CRNDEP is likely capable of RNA binding. All these results may contribute to a better understanding of the regulatory mechanisms the *CRNDE* gene expression products are involved in, as well as improvements in ovarian cancer screening and diagnosis. Ultimately, this may also lead to the development of less aggravating and more effective methods for the future molecular therapy of this disease.

## Figures and Tables

**Figure 1 ijms-25-04381-f001:**
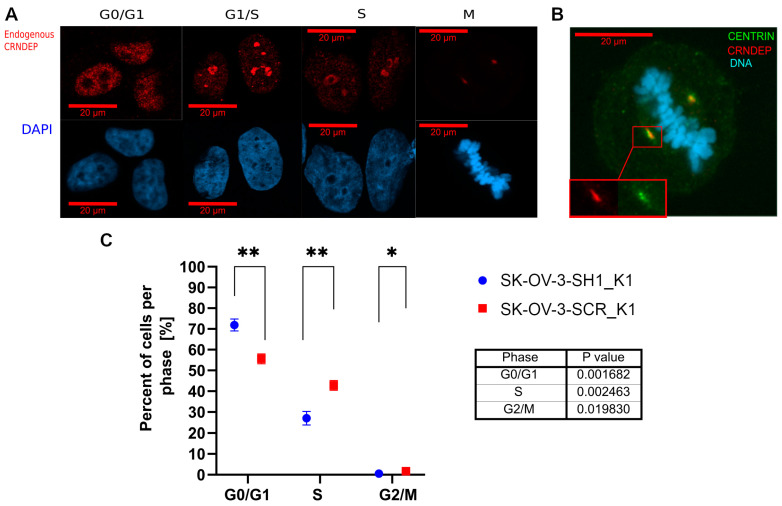
CRNDEP and the cell cycle. Changes in CRNDEP localization depending on the cell cycle phase were evaluated in HeLa cells under a fluorescence microscope (**A**,**B**), using the primary α-CRNDEP antibody detected with the secondary antibody conjugated with the Alexa Fluor 647 dye (red). Cell nuclei were stained blue with DAPI. (**A**) CRNDEP localization in different cell cycle phases. (**B**) Co-localization of CRNDEP (red) with centrin (centrosome marker; green) in the M phase. (**C**) Distribution of SK-OV-3 cells with (SK-OV-3-SH1_K1) and without (SK-OV-3-SCR_K1) CRNDE(P) silencing in different cell cycle phases. The analysis was performed using flow cytometry, based on the intensity of cell nuclei staining with PI. Each point in the graph represents mean value obtained from 3 independent measurements, which is presented here as the percentage of cells in the given cell cycle phase. PI—propidium iodide. *—0.01 ≤ *p*-value < 0.05; **—0.001 ≤ *p*-value < 0.01.

**Figure 2 ijms-25-04381-f002:**
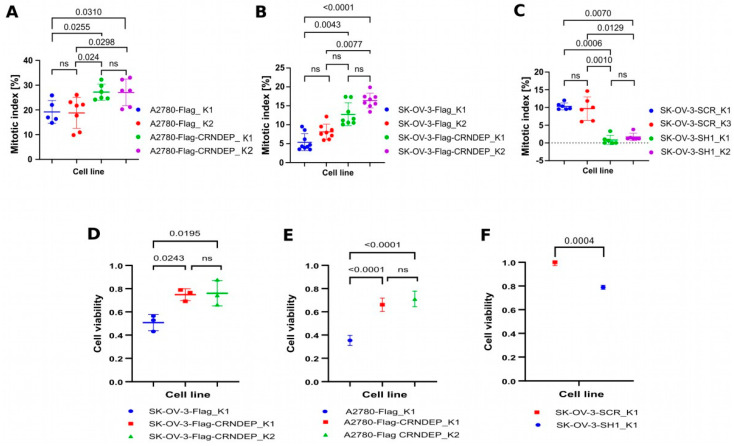
Impact of CRNDE(P) expression on cell proliferation and metabolic activity. (**A**–**C**) Mitotic index evaluation in two ovarian cancer cell lines with CRNDE(P) overexpression, A2780 (**A**) and SK-OV-3 (**B**), or CRNDE(P) silencing in SK-OV-3 cells (**C**). (**D**–**F**): MTT test for metabolic activity assessment in two ovarian cancer cell lines with CRNDE(P) overexpression, SK-OV-3 (**D**) and A2780 (**E**), or CRNDE(P) silencing in SK-OV-3 cells (**F**). In each plot, the control cell lines (Flag_K1/2, SCR_K1/3) are located on the left. ns—non-significant result.

**Figure 3 ijms-25-04381-f003:**
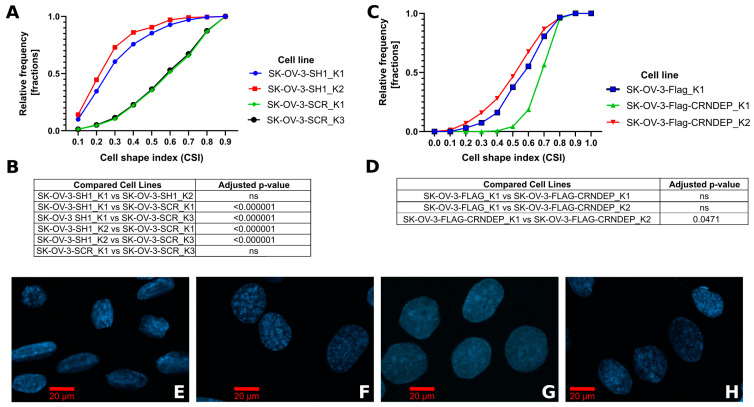
Cell shape index (CSI) assessment. (**A**) Cumulative distribution of CSI coefficient values in SK-OV-3 cells with CRNDE(P) silencing (SK-OV-3-SH1_K1/2) compared to the control SK-OV-3 cells (SK-OV-3-SCR_K1/3). (**B**) Statistical analysis of differences in the distribution of CSI coefficient values in SK-OV-3 cells with CRNDE(P) silencing. (**C**) Cumulative distribution of CSI coefficient values in SK-OV-3 cells with CRNDE(P) overexpression (SK-OV-3-Flag-CRNDEP_K1/2) compared to the control cell line (SK-OV-3-Flag_K1). (**D**) Statistical analysis of differences in the distribution of CSI coefficient values in cells with CRNDE(P) overexpression. (**E**–**H**): Representative microscopic images of the cell lines (nuclei stained with DAPI): SK-OV-3-SH1_K1 (**E**), SK-OV-3-SCR_K1 (**F**), SK-OV-3-Flag-CRNDEP_K1 (**G**), SK-OV-3-Flag_K1 (**H**); ns—non-significant result.

**Figure 4 ijms-25-04381-f004:**
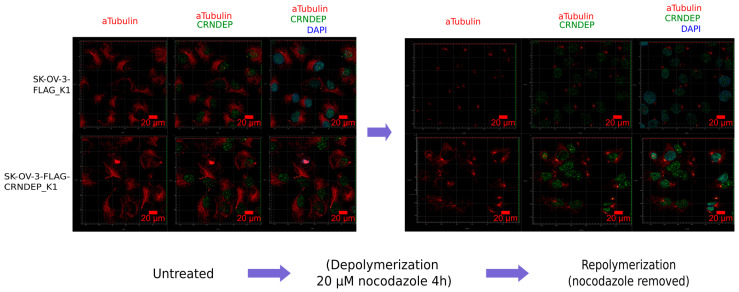
Microtubular cytoskeleton remodeling in SK-OV-3 cells with CRNDE(P) overexpression. In SK-OV-3 cells with overexpression of Flag-CRNDEP fusion protein (SK-OV-3-Flag-CRNDEP_K1), the rate of microtubule repolimerization after nocodazole withdrawal from the culture medium was higher than in the control cell line (SK-OV-3-Flag_K1). aTubulin—alpha-tubulin.

**Figure 5 ijms-25-04381-f005:**
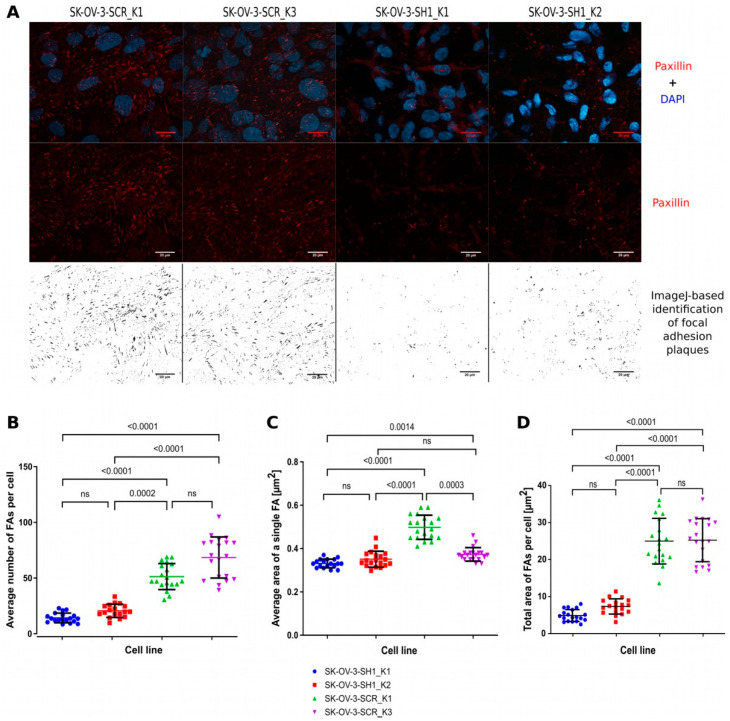
Evaluation of cell–substrate adhesion changes caused by CRNDE(P) silencing in SK-OV-3 cells. In figure (**A**), a representative microscopic image of paxillin, a marker of focal adhesion plaques (FAs), in SK-OV-3 cell with (SK-OV-3-SH1_K1/2) and without (SK-OV-3-SCR_K1/3) CRNDE(P) knockdown is shown. Each scale bar is 20 µm long. Underneath, plots portraying differences in the average number (**B**), average area (**C**), and total area (**D**) of FAs, supplemented with the statistical analysis results, are presented; ns—non-significant result.

**Figure 6 ijms-25-04381-f006:**
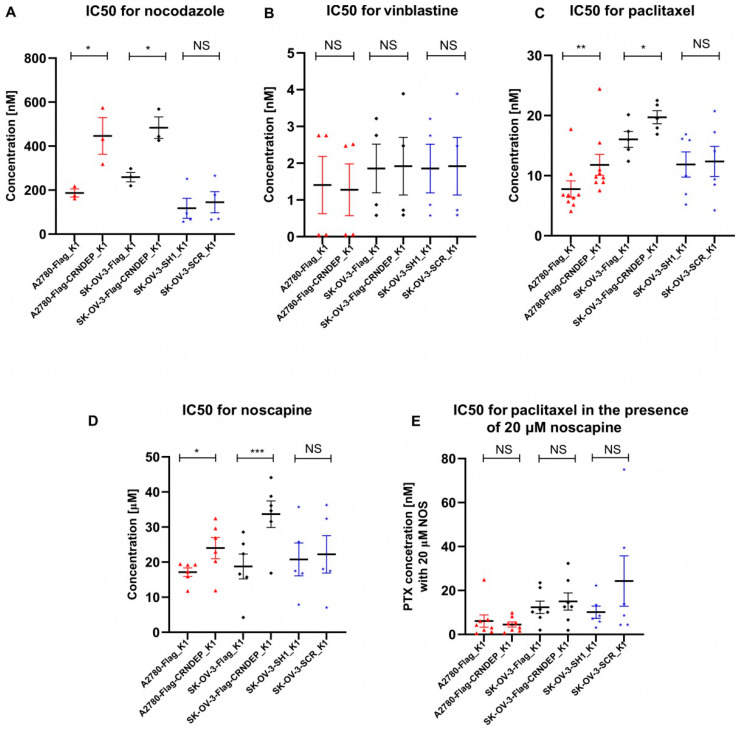
Influence of CRNDE(P) expression on sensitivity of ovarian cancer cell lines to microtubule-targeted cytostatic agents: nocodazole (**A**), vinblastine (**B**), paclitaxel (**C**), noscapine (**D**), and paclitaxel combined with 20 μM noscapine (**E**). Horizontal bars represent mean IC50 values; vertical bars represent standard error of the mean (SEM) values. Each point on the plots corresponds to a IC50 value obtained in a single experiment. *—0.01 ≤ *p*-value < 0.05; **—0.001 ≤ *p*-value < 0.01; ***—*p*-value < 0.001. NS—non-significant result.

**Figure 7 ijms-25-04381-f007:**
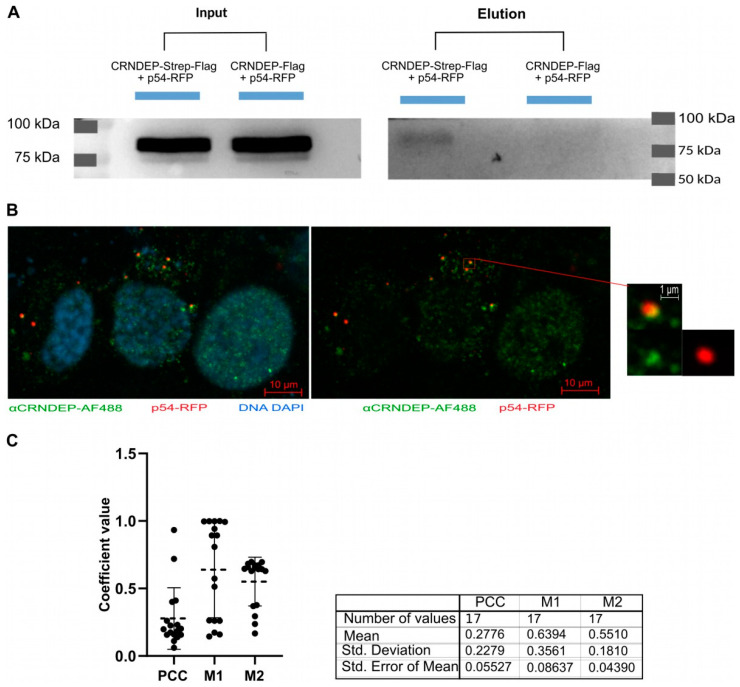
Interaction of CRNDEP with p54. This interaction was first confirmed in vitro by co-immunoprecipitation of two transgenes ectopically expressed in HeLa cells (CRNDEP fused to the Strep and Flag tags and p54 conjugated with RFP) on a magnetic resin coated with streptavidin, followed by Western blotting with the anti-RFP antibody (**A**). This result was further supported by in situ co-localization experiments in SK-OV-3 cells stably expressing the Flag-CRNDEP fusion protein with concomitant ectopic expression of p54-RFP (**B**). The latter analysis was also supplemented with a statistical evaluation of Pearson’s correlation and Manders’ overlap coefficients (M1 and M2) (**C**), which proved that both proteins in fact interact in the cell, though this interaction seems to be partial given the relative low values of all the coefficients. RFP—red fluorescent protein; PCC—Pearson’s correlation coefficient; αCRNDEP-AF488—the primary rabbit anti-CRNDEP antibody detected with the secondary anti-rabbit antibody conjugated with the Alexa Fluor 488 fluorescent dye (green).

**Figure 8 ijms-25-04381-f008:**
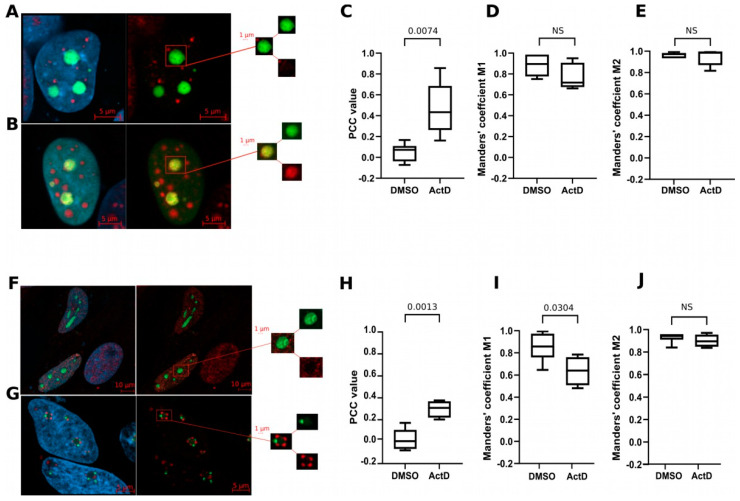
Co-localization of CRNDEP with nucleolin and fibrillarin in the presence of actinomycin D. Results are presented for HeLa cells transfected with plasmids expressing nucleolin (upper panel) or fibrillarin (lower panel), both in a fusion with GFP. In each experiment, endogenous CRNDEP was detected using the secondary anti-rabbit antibody (against the primary rabbit anti-CRNDEP antibody) conjugated with the Alexa Fluor 647 dye (red). Twenty-four h after transfection, the control cells were treated with DMSO (**A**,**F**) and the test cells with actinomycin D (its stock was dissolved in DMSO) (**B**,**G**). Cell nuclei were stained with DAPI (blue). Co-localization results were calculated for 20 cells from each group, using the Pearson’s correlation coefficient (**C**,**H**), as well as Manders’ overlap coefficients M1 and M2 (**D**,**E**,**I**,**J**). Each plot is additionally supplemented with the statistical result of the Welch’s *t*-test. NS—non-significant result; PCC—Pearson’s correlation coefficient; ActD—actinomycin D; GFP—green fluorescent protein.

## Data Availability

All data are available in the main text or the Appendix A.

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
