# Peer review of "A Multi-Faceted Analysis Showing CRNDE Transcripts and a Recently Confirmed Micropeptide as Important Players in Ovarian Carcinogenesis"

_ijms, 2024, doi:10.3390/ijms25084381_

Round 1

Reviewer 1 Report (New Reviewer)

Comments and Suggestions for Authors

This manuscript is the corresponding author’s third paper regarding CRNDE. This gene is generally considered an oncogenic lncRNA in solid tumors and hematologic cancers. The authors discovered that CRNDE encodes a nuclear micropeptide CRNDEP in highly proliferating tissues such as ovarian cancer cells, endometrium in proliferative phase, tonsillar germinal center, intestinal crypt, etc. In this sequel 3, the authors explored the functions of the short peptide. They found CRNDEP localized in centrosome during mitosis and interfered microtubule dynamics, indicating its role in the resistance of microtubule stabilizers. The authors also discovered the micropeptide localized in nucleoli and interacted with RNA helicase, indicating its role in RNA metabolism.

 The authors used overexpressed and KO cell lines. lncRNA and micropeptide are encoded by different transcripts as described in the introduction. However, both CRNDE lncRNA and micropeptide expression levels were affected in these models as shown in Figures S1-S3. Is it possible to modify the expression of the two variants respectively and study their functions independently?

The PDF file is in a rather odd format. In addition, figures 1, 2, 5 and 6 are missing. Line 185 on page 8 is covered by Fig. legend.

The discussion is too long.

Author Response

Reviewer 2 Report (New Reviewer)

Comments and Suggestions for Authors

In their paper Balcerak et al. focused on the role of CRNDE transcripts and the corresponding micropeptide (CRNDEP) in ovarian cancer. The authors explored the impact of CRNDE as oncogene that can affect carcinogenesis at different levels spacing from DNA transcription and replication, and cell proliferation as well as cell cycle progression. In particular, they focused on two ovarian carcinoma cell lines (SK-OV-3 and A2780) expressing CRNDEP fused to a double flag tag as well as CRNDE knockdown models. Such an approach allowed the authors to explore both CRNDEP localization along different cell cycle phases and sensitivity to microtubule-targeted agents.

The manuscript presents several major criticisms  that need to be addressed reported as follows.

·       ·     CRNDE abbreviation is not explicit, the full name at the beginning of the manuscript should be included.

·     The authors performed their studies on SK-OV-3 and A2780 cell lines, used in the study as cell models of ovarian cancer. However, the choice of these cell lines, their similarities and differences, as well as their molecular features should be clarified by authors to justify their use in the study.

·       ·     Fig.1, Fig. 2; Fig. 5; Fig. 6 are missing in the text, making hard the evaluation and judgment of the whole manuscript.

·       ·     Microscope’s magnification or scale bar should be reported in the legend of Figure 3 and Figure 4.

·      ·     At page 6, authors performed their analyses by using nocodazole to analyze the rate of microtubule repolymerization. Differently, the description of this molecule is reported at page 8 (line 179-180) together with other microtubule targeting agents. This description should be brought forward at page 6.

·      ·     All analyses were performed without any statistics. Please, add this aspect in the manuscript specifying the statistical test used for data analysis.

·      ·     Discussion is too long, it should be shortened. For example, lines 384-401 should be moved in the introduction section highlighting the role of p53  and its possible relationship with CRDNE expression.

Comments on the Quality of English Language

Comments on the Quality of English Language

The manuscript needs a careful reading for minor english corrections such as - ......”HeLa cell”... (line 261). .... genes encoding the SLUG, TWIST and vimentin proteins (SNAI2, TWIST1, 480 VIM) “had significantly” (lines 480-481).

Round 2

Reviewer 2 Report (New Reviewer)

Comments and Suggestions for Authors

Tha manuscript was ameliorated by authors and can be accepted for publication.

Comments on the Quality of English Language

Minor editing of English language is required.

This manuscript is a resubmission of an earlier submission. The following is a list of the peer review reports and author responses from that submission.

Round 1

Reviewer 1 Report

Comments and Suggestions for Authors

The MS entitled “Multi-Faceted Analysis Showing CRNDE Gene, Transcripts and the Lately Confirmed Micropeptide as Key Players in Ovarian Carcinogenesis” describes molecular studies and the effects mediated by a CRNDE peptide. While new results are provided, many questions could be addressed. 

  1. Does CRNDE(P) and CRNDEP mean the same? If it is, please use one.
  2. A “key player” is too much overestimated. You should use a more conservative indication. The dominance of the CRNDEP role is not clear from the MS.
  3. With only three references, the Introduction looks odd. You should expand this section with a more detailed CRNDE/CRNDEP description or provide similar examples.
  4. For each statistical significance estimation and p-value, a test type should be provided.
  5. Cells could not be more or less viable. Please remove this inaccuracy.
  6. 3 does not have error bars. Was it a single experiment in each case?
  7. What is the evidence that the observed knockout and CRNDEP hyperproduction effects are not due to off-targeting?
  8. 10 quality is bad.
  9. The discussion section also looks very odd, resembling a review. Maybe it is a good idea to transfer something in the introduction.

Comments on the Quality of English Language

Generally, English is fine.

Reviewer 2 Report

Comments and Suggestions for Authors

comment 1

explain all initials in the first appearance

comment 2

Our previous paper is the only one to report CRNDE as a micropeptide-coding gene. change reporting

comment 3

Fig 1 and 4

use the same appearance (choose with or without bar)

comment 4

put the pictures with better focus

comment 5

in page 10 down regulated gene numbers should be (N=29) instead of (29)

comment 6

microscopic pictures in figure 8 are good

comment 7

add microscopic picture of cell shape and size change with the nuclear size whether polyploidy or not

if nuclear size increase add the evidence of polypidy in cells with changed size 

comment 8 

well designed reverse genetics

Reviewer 3 Report

Comments and Suggestions for Authors

The authors investigate the function of the microprotein encoded by CRNDE in relation to its carcinogenesis. For this purpose, the ovarian cancer cell lines SK-OV-3 and A2780 are genetically modified in such a way that the studies can be carried out on knockdown, flag-tag labeled positive and down-regulated cell lines. This made it possible to detect one locus of the expression of the microprotein in the centrosomes. Furthermore, the microprotein also appears to have functions in the nucleolus, where it is presumably involved in nucleic acid synthesis. It also appears to have an influence on cell-substrate adhesion, microtubule dynamics and cell shape. The authors identify stress granules, chaperones, microtubules, chromatin, focal adhesion and p54 as interaction partners of the microprotein. With CRNDE expression, an increase in the mitotic index and viability can be detected. At the same time, higher expression seems to be associated with less aggressive forms of ovarian cancer. In addition, CRNDE expression promotes resistance to cytostatic drugs such as nocodazole and paclitaxel. In this regard, the authors discuss a combination treatment of these cytostatica in order to overcome their resistance. From these results, the authors conclude that the microprotein CRNDEP could be a negative prognostic marker for ovarian cancer. This hypothesis was tested in 227 patients with ovarian cancer.

The authors present a very detailed work that represents very deep and comprehensive experimental work. With 39 pages of manuscript and 46 pages of additional supplementary material, the volume clearly exceeds the scope of a scientific article. In terms of content, several manuscripts can be generated from this material. The manuscript should urgently be reduced to essential components so that the information can be grasped by the reader. However, despite all the abundance of experiments, the essential authentication of the cell lines involved, such as through STR marker analyses, is missing. This is particularly important because the authors discuss many results against the background of a p53 mutation of SK-OV3. In the discussion, the p53 mutation of SK-OV3 is often discussed in detail in relation to the results of other research groups. Here the reviewer is missing the reference to A2780 and the comparison to its data.

This manuscript should be published when revised.

Round 2

Reviewer 1 Report

Comments and Suggestions for Authors

My issues were resolved, generally. However, the PDF attached looks odd. It looks like there is some kind of problem with line indentation.

Reviewer 3 Report

Comments and Suggestions for Authors

The authors investigate the function of the microprotein encoded by CRNDE in relation to its carcinogenesis. For this purpose, the ovarian cancer cell lines SK-OV-3 and A2780 are genetically modified in such a way that the studies can be carried out on knockdown, flag-tag labeled positive and down-regulated cell lines. This made it possible to detect one locus of the expression of the microprotein in the centrosomes. Furthermore, the microprotein also appears to have functions in the nucleolus, where it is presumably involved in nucleic acid synthesis. It also appears to have an influence on cell-substrate adhesion, microtubule dynamics and cell shape. The authors identify stress granules, chaperones, microtubules, chromatin, focal adhesion and p54 as interaction partners of the microprotein. With CRNDE expression, an increase in the mitotic index and viability can be detected. At the same time, higher expression seems to be associated with less aggressive forms of ovarian cancer. In addition, CRNDE expression promotes resistance to cytostatic drugs such as nocodazole and paclitaxel. In this regard, the authors discuss a combination treatment of these cytostatica in order to overcome their resistance. From these results, the authors conclude that the microprotein CRNDEP could be a negative prognostic marker for ovarian cancer. This hypothesis was tested in 227 patients with ovarian cancer.

The authors still present a very detailed work that represents very deep and comprehensive experimental work. With 42 pages of manuscript the volume clearly exceeds the scope of a scientific article and the extent before revision. The authors did not follow my advice to shorten the manuscript. Instead, they justify the need for this form because it is only the second paper dealing with CRNDE. Furthermore, the authors do not comply with the request to authenticate the cell lines used. This is absolutely necessary because we know from the literature that around 50% of research on cell lines is falsified with contamination. Since the authors do not want to prove cell line authentication as an urgently necessary condition for good scientific work, I have to reject the manuscript.